# Alluvial Artisanal and Small-Scale Mining in A River Stream—Rutsiro Case Study (Rwanda)

## Jan Macháček

Department of Social Geography and Regional Development, Faculty of Science, University of Ostrava, Chittussiho 10, 710 00 Ostrava, Czech Republic; jan.machacek@osu.cz; Tel.: +420-733-756-592

**Abstract:** Artisanal and small-scale mining is a significant economic sector in Rwanda. Mining activities often use a watercourse, in which secondary extraction takes place and minerals are washed. Mining thus greatly affects the geomorphological conditions in the area. The aim of this paper is a digest of environmental impacts of alluvial artisanal and small-scale mining with a focus on anthropogenic influences on topography with regard to the methods used in raw material mining. The author draws on a case study from the mining site of Rutsiro district in Rwanda. Main findings of alluvial artisanal mining in a riverscape are changes in landscape structure, deforestation, intensification of geomorphological processes, new relief shapes (suffosion depressions, check dams, gravel benches, anthropogenic channels) and hydrological river regime, chemical pollution of soil and watercourses. Artisanal and small-scale mining may lead to a significant change and acceleration of fluvial processes. This paper covers a broad understanding of environmental impacts of alluvial artisanal and small-scale mining with a focus on anthropogenic influencing.

**Keywords:** human impact; artisanal and small-scale mining; river; minerals; Rwanda

## 1. Introduction

Artisanal and small-scale mining (ASM) is one of the most important rural non-agricultural activities in the developing world. It is an important source of employment and income for dozens of millions of people and brings economic benefits to other millions who are not directly involved in ASM. According to some estimates, six secondary jobs are created for each job in the ASM sector [1]. The most discussed topics in ASM include socio-economic aspects of mining [2–7], child labor [8,9], women's labor [10–15] and the role of minerals in armed conflicts [16,17]. Less attention is then paid to the actual environmental impacts of mining and their typology [18], with these topics being only marginally addressed in scientific papers. The environmental impacts of ASM in the Great Lakes Region (the author uses the geography division similar to Mpangala [19] or Schütte et al. [20], that includes Uganda, Kenya, Tanzania, Burundi, Rwanda, and the eastern part of the Democratic Republic of Congo, or the province of North Kivu, South Kivu, and Katanga in the Great Lakes Region—Figure 1) and Africa are grouped into four categories. The four categories are: (1) Changes in landscape structure—deforestation (primary and secondary) and land cover change. (2) Influence of geomorphological processes—weathering, mass movements, fluvial processes, aeolian processes, creation of new anthropogenic forms. (3) Influence of hydrological regime—water contamination, sedimentation of water stream. (4) Influence on fertility of soil–Soil contamination, high dustiness, land use change [18].

ASM in the flood plain has direct negative impacts on the stream's habitat. The most significant impact is streambed stability and composition, channel shape, turbidity, velocity, water depth, the amount of woody material in the channel [21–24]. According to the Communities and Small-scale Mining Organization, the most visible impact of ASM on the environment is deforestation and

destruction of vegetation in mining areas [25]. Maponga et al. [26], Kanyamibwa [27], Plumptre [28], Karamage [29], Beyene [30] also deal with land cover change and vegetation removal in the Great Lakes Region as a result of ASM.

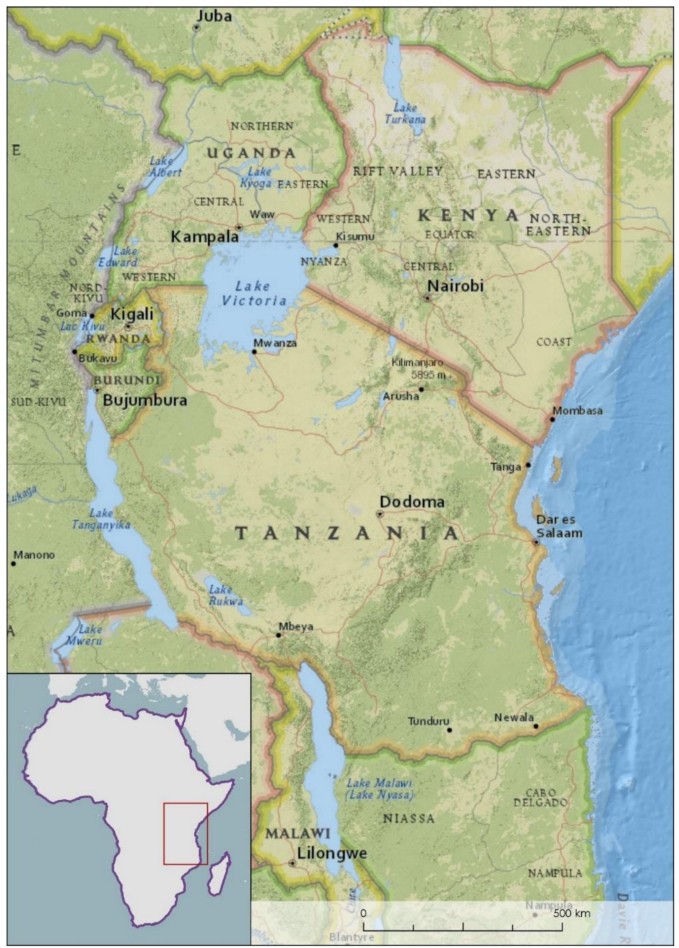

**Figure 1.** African Great Lakes Region. Author's own format [31].

Most papers deal with this in connection with gold mining and mercury contamination in mining areas. Mercury enters water [32–34], soil [34–36], and air [32,36] during gold mining from ores through amalgamation [37]. For example, Lacerda [38] notes that more than 24 tons of mercury were discharged into the atmosphere at the Lake Victoria Goldfields Site in Tanzania. At the same location, research was carried out by Ikingura et al., who measured extreme mercury concentration in the soil [39]. According to a number of research works at mining sites, mercury can also enter the human body, as argued by, for example, Sakoane [40], who links the spread of malaria to higher mercury concentrations in the soil. The impact of heavy metals (excluding mercury) on the environment during the extraction of minerals is addressed by Diogo et al. [41] and Pourret et al. [42] who deal directly with ASM. Lead poisoning, especially in child labor, is the subject of the report by the WHO entitled Artisanal and Small-Scale Gold Mining and Health [43]. Mossa and James [44], Knox [45], Wolfenden and Lewin [46], Graf et al. [47] deal with the impact of heavy metals on watercourses in mining in general. Alluvial ASM has an effect on the watercourse and its immediate surroundings, while some problems can be eliminated, such as the impact on agricultural activities and subsequent soil erosion [48–50], which has an impact on vegetation cover [34,51,52] and biogeographic conditions. Vegetation layers and biogeographical conditions can be studied using several models [53,54], which, however, is not possible in informal ASM conditions.

ASM takes place in more remote, mostly poorer areas, and therefore those sites may experience uncontrollable illegal mining and smuggling of minerals from other areas or countries. In poorer rural areas, the only source of income apart from agriculture is often the extraction of minerals. Mining in these areas is more profitable than agricultural activities, which has in particular a self-sufficiency function. Due to the large population size, high population density of the Great Lakes Region, and a greater dependence of the population on quality soil and water, environmental aspects play an important role in the life of the local population. A study conducted in Rwanda by Cook et al. [55] documents that in mineral-rich areas, mining is the dominant source of income for most workers and has become increasingly important in the last ten years. Most miners claim that mining offers an above-standard source of income. They can earn twice as much money by mining than in agriculture, thus these activities tend to develop, while natural conditions in the affected areas worsen.

Mining of minerals is one of the most important industrial activities in Rwanda, and the anthropogenic influence on the relief is an integral part of it. ASM plays a significant role in mineral mining, in which water is used for washing process. Given the presence of minerals in the alluviums, ASM has an impact on the floodplain. Mining in alluvial sediments leads not only to a change in the hydrological regime of the river, but also to the deforestation and degradation of arable land near watercourses. In the context of water resource management, the water used in mining is polluted both chemically and physically, with large amounts of suspended sediment appearing in the water.

ASM in Rwanda is characterized by hard rock mining and shallow alluvial mining. Hard rock mining is mining by the extraction of material from mineral veins. Shallow alluvial mining is mining from alluvial sediments, where material is extracted followed by the "dig and wash" technique [18,56]. By using these two types of mining, miners destabilize the river channel by bedrock mining on the riverbanks. An ore-bearing rock washing process occurring in the river has an additional effect on drinking water quality. Mining activities result in the degradation of riverbeds and riverbanks, and changes in water flow. The result of these activities can be observed, among other things, in the lower reaches of rivers, where sedimentation of suspended sediments, widening of the watercourse and in some cases also a reduction in water flow and drying of the riverbed occur. The mining in the alluvium then significantly contributes to the acceleration of natural geomorphological processes. The primary objective is to document anthropogenic impacts of the alluvial ASM of mineral raw materials, especially 3T minerals (tin, tantalum, tungsten) on a fluvial landscape in Rwanda. The paper aims to bring to ASM issues expanding knowledge about alluvial mining. Based on the previously created typology of environmental impacts in Macháček (2019), alluvial mining will be incorporated in the noted typology and new relief shapes formed during this type of mining will be defined. Furthermore, the impact of alluvial ASM on the riverscape will be analyzed in connection with the degradation of arable land, which is especially in developing countries an important source of livelihood for the local population.

## 2. Anthropogenic Impacts of ASM On Geomorphological Processes

Unlike industrial mining, whose phases of landscape damage are known, and there are ways to eliminate these risks, in the field of ASM these ways and procedures are not entirely clear. In the initial phases of industrial mining, the landscape in the mining areas changes very quickly. When mining reaches its peak, activities are reduced, and after the deposits are depleted, the mine is closed and subsequent reclamation takes place. In the case of ASM, in most cases there is no mining plan or subsequent reclamation plan. This is an unplanned and often informal activity without detailed plans, and therefore it is not possible to estimate the direction in which both the actual mining and the impact on the environment will be heading. In areas affected by ASM, as a result of mining, both the rock environment and soil properties may start changing. If subsequent reclamation is not planned at the beginning of mining activities, it can happen spontaneously, unfortunately, with the worst results. In abandoned mines, where no subsequent reclamation has taken place, spontaneous movements can occur, such as landslides and rock falls. Unsecured mines also pose a major risk to the local population and livestock.

ASM can either initiate new or modify (accelerate or slow down) geomorphological processes that have already occurred naturally. These dynamic processes are influenced by the topography of the relief, the soil properties and the rock composition. Thus, in these cases, anthropogenic activities may lead to faster re-degradation or aggradation of the relief landforms.

All types of mineral extraction through either ASM or large-scale mining, involve influencing of geomorphological and geological structures that directly or indirectly affect geomorphological processes. According to Jones [57], we can divide this anthropogenic influence into three categories: human-made relief, human-induced relief, and human-modified relief. Although Jones [57] divided the anthropogenic influence of relief into several categories, he also did not precisely define geomorphological processes closely associated with ASM and occurring only in this issue. One of the few studies that strives to define individual characteristics is the study by Byizigiro et al. [58], who hypothetically describes the geomorphological processes characterizing ASM. Byizigiro bases his description on the division of relief according to Jones (2001) and, similar to the author of this paper, deals with the area of Great Lakes Region, in which he worked between 2012 and 2015.

According to Jones [57], the human-made relief is intentionally created for a specific purpose, e.g., to remove overlying rock material, so that in the case of mining, mineralized material could be used. Other landforms directly related to ASM include partial excavations in various parts of the mining area, which serve as exploration wells. Another phenomenon is the mining pits located on plains or slopes, which are artificially created depressions in the ground used mainly for illegal mining. Some mining sites on the slopes are disturbed by shafts, which may collapse during or after mining, creating surface depressions. Another important element is the mining wall itself, which is already prone to further erosion due to natural processes. The upper soil is exposed down to mineralized rock, while the no longer usable soil is deposited in the form of barren rock near the mining site. These newly created elements are susceptible to other processes, such as erosion or collapse of the accumulated material.

The human-induced relief is a product of natural processes at a place and time and is completely dependent on anthropogenic activity. In ASM areas, geomorphological processes take place in setting pits, mining pits, and barren rock or in their vicinity. These elements were created primarily by human activity and are features of human-induced relief. The result of such disruption of the relief is wash and other erosion rills, which later expand, thus resulting in even greater disruption of the already affected relief. These processes can subsequently result in formation of badlands [58]. One disruption in a slope where intense ASM takes place can lead to subsequent destabilization of the upper part of the slope, resulting in cracks. In this way, the strength of the slope is reduced. If the cracks are filled with water, which occurs in the form of precipitation, weakened rock blocks can collapse and slide. These weakened zones often provide an area for further mass movements at the top of the slope. The landslides themselves then often adversely change the slope geometry and lead to further instability of the rock environment. Less obvious and more hidden are the movements inside the rock environment. The strength of these deposits is significantly affected by internal displacements, which can be converted at the points of movement by pressure from relatively loose material to firm parts on which movement takes place. Byizigiro et al. [58] found that this effect was stronger and more effective in the walls of mining pits parallel to the shale layers, which are largely eroded by landslides.

We refer to the human-modified relief when the extent or speed of geomorphological processes is changed by human activity [57]. The main mechanism that triggers the formation of these processes is the change in hydrological balance. This change can occur due to a change in the soil cover, exploratory wells or over-extraction, which disrupts the groundwater supply.

Alluvial ASM thus affects all three types of relief defined by Jones [57]. There is a broad consensus among experts on the extent of geomorphological processes influenced by (accelerated, slowed down) ASM. The most important is the influence of erosion processes. However, in the areas of ASM, erosion processes are very poorly monitored and the conditions for reducing the consequences of process acceleration due to anthropogenic influence have not yet been created. At the same time, accelerated fluvial erosion fundamentally affects the quality of natural resources, the loss of arable

land and the environment degradation. Erosion activities are manifested mainly by the loss of topsoil and the subsequent sedimentation of watercourses in the mining area [49,51,59–61]. This process can further lead to a change in the riverbed (caused by physical disturbance of the banks and vegetation) and the destruction of vegetation [62]. According to Li [63], the sites with ongoing ASM can constitute extremely difficult conditions for subsequent reclamation. The main reasons include insufficient preparation of reclamation processes before the actual start of mining. Insufficient control by government authorities is also connected with this fact, as ASM often takes place in remote areas covering a large area. The major environmental issues related to the minerals mining using ASM include low-formalized activity, unequal access to ASM and large-scale mining miners, disadvantaging of ASM miners, insufficient government access to ASM, and little attention to environmental issues in general. These issues associated with ASM are then insufficiently coordinated with national and international mining policies, resulting in insufficient regulation of the ASM sector [64]. The problem is significantly exacerbated in gold mining, where mercury is used. Artisanal and small-scale gold mining with mercury is mainly carried out in the areas of East and Southeast Asia, South America, and Sub-Saharan Africa, and many of them are located in floodplains [65]. The exact and complete impact of ASM on floodplains is not yet known [64].

During ASM, the natural structure of rocks and soils is disrupted, and they become crumbled, separated, and their composition is changed. As a result of the movement of materials, an anthropogenic rock weathering crust is formed. This significantly changes the geodynamic, geochemical, geothermal and gravitational situations. There are also major changes in the chemical-mineral, granulometric and physical-mechanical properties of rocks and soils, especially in terms of their disintegration. Rarely we find secondary strengthened materials in anthropogenic rock weathering crusts [66].

The degree of influence of mining on the landscape depends on the type of ASM and the extent of mining activities. Individual types of ASM can be defined according to technological procedures and their effect on geomorphological processes. Technological procedures used in ASM are mainly associated with surface and subsurface mining, which takes place in rock masses on areas with different slope inclines or in alluvial plains. The methods used to extract the material can be divided into three categories: shallow alluvial mining, deep alluvial mining, and hard rock mining [18].

Extraction methods and further processing are based on relatively simple methods. These techniques have been used since the 19th century and are mostly based on rock flushing and capturing extracted material that is heavier than the original rock [4]. In areas where ASM is widespread, the three most common approaches toward mineral extraction are assumed. The first extraction method is simple sluicing, which is the easiest way to separate rock from mineral, then ground sluicing is a method that is used in mines without the use of advanced technologies and in often illegal mines, and finally, the third applied method, hydraulic mining, is the extraction of minerals from the rock supplied with enough water and available technological equipment [18].

The output of 3T mineral extraction is the so-called concentrate, which is obtained by the two most used techniques in ASM, that is, gravity concentration and comminution [67].

- Gravity concentration is a process used to concentrate a mineral of interest. The technique uses the physical properties of the minerals and rocks in which the minerals occur. Grains or larger pieces of mineral are moved by gravity force to the bottom of the vessel, in which this process occurs. The oldest technique, already known from gold mining in the Middle Ages, is panning. Circular and retrograde movements of ore and water in a pan cause the ore to stratify, as heavy minerals settle to the bottom of the pan, allowing lighter pieces of barren rock to sluice from the upper part. Panning is a basic tool to collect minerals from alluvial deposits and also from high-quality primary ore [68].

- Comminution is a technical term used to describe mechanical disintegration of a rock, which is done by crushing (coarse) and grinding (fine) or by simply breaking up of a lump of soil or clay materials. This is the processing of already extracted material, which is deposited in the form of barren rock [67]. The barren rock generated by these activities are in many cases commonly

discharged and moved to the surroundings of the mining area due to insufficient legislative standards or a lack of storage facilities. Therefore, these technological processes have a great influence on geomorphological processes and on the environment [69]. Table 1 illustrates the influence of geomorphological processes by individual material mining and extraction methods.

**Table 1.** Influence of geomorphological processes based on the chosen mining method and material extraction [18].

| Process Characteristics | Process Influenced (x = yes, 0 = no) | | | | | Form Influenced |
|---|---|---|---|---|---|---|
| | Rock Breaking, Weathering | Mass Movements | Fluvial Processes | Cryogenic Processes | Aeolian Processes | |
| Mining method | | | | | | |
| Shallow alluvial mining | x | 0 | x | 0 | x | alluvial plain, levee |
| Deep alluvial mining | x | 0 | x | 0 | x | alluvial plain, levee |
| Hard rock | x | x | x | 0 | x | cret, fork, back, influencing entire solid rock mass |
| Extraction method | | | | | | |
| Simple sluicing | x | 0 | x | 0 | x | alluvial plain, levee |
| Ground sluicing | x | 0 | x | 0 | 0 | cret, fork, back, alluvial plain, levee |
| Hydraulic mining | x | x | x | 0 | x | cret, fork, back, alluvial plain, levee |
| Gravity concentration | x | x | x | 0 | x | cret, fork, back |
| Comminution | x | x | x | 0 | x | cret, fork, back |

The relationship between anthropogenic activities and sedimentation can be well illustrated by mineral material mining. Mining sediment not only provides evidence of fluvial processes, but also demonstrates the impact of human activities on environmental change. All mining and processing of minerals produces some waste, which is either separated with help of watercourses, or is intentionally added to them, or eventually enters watercourses through natural processes. According to Kondolf et al. [70], we distinguish two types of sediment transfer. Active transformation, where the fluvial system is affected by the active addition of waste and passive dispersal, where parts of the sediment in a watercourse are mixed with natural sediments without causing a significant change in the watercourse morphology [71]. However, the ability to distinguish mining sediment from common sediments in a river floodplain can be problematic. Extraction is often accompanied by other territorial and river hydraulic changes, such as agriculture, cultural intensification, deforestation or the construction of dams. This may indirectly increase sedimentation from other sources [44]. Other factors affecting the landscape near alluviums include the number of miners, mining methods, degree of mechanization and other factors [72,73]. A significant phenomenon is episodic sedimentation. As claimed by Kondolf and Piégay [70] episodic sedimentation generated by human land-use change, such as deforestation, ploughing for agriculture and mining, may be sufficiently severe to cause channel and floodplain aggradation that is preserved in the alluvial record. Aggradation is often followed by a

period of recovery in response to relaxation of the causative factors (reforestation, cessation of mining, etc.) and channel incision that leaves sediment stored on floodplains.

Fluvial processes are associated with running water activities, which is the main erosion factor. The development of the landscape and the river network is dependent on fluvial processes. A fundamental factor for influencing fluvial processes is the disturbance of vegetation cover in the spring sections of watercourses [74]. An important factor associated with fluvial processes is deforestation. Deforestation can occur as a result of fires, forest clear-cutting aiming at expansion of arable land, or as a result of mineral raw material mining [18,75,76]. The combination of these phenomena is a common cause of influencing fluvial processes in developing countries. Vegetation cover, especially forest stands, plays an important regulatory function, retaining some of the water that falls in the form of precipitation, thus slowing down evaporation. Soil erosion entails soil degradation, and this leads to a reduction in its fertility. In addition to fluvial erosion, organic matter loss, salination, and chemical contamination also contribute to soil degradation. The intensity of fluvial processes can be expressed numerically by the volume of eroded and transported material by watercourses. This material is in the form of suspended sediment, which, however, demonstrates the speed of natural and anthropogenic processes together [74].

Water turbidity and sediment vary according to the level of pollution, which increases with distance and number of tributaries and activities on the river. The water color also varies according to the soil type and use type, e.g.,: the water is more or less clear at some distance from its source before being infused with sediments, dark brown from intensive mining activities, and dark grey or black due to organic matter being transported from marshland and steep slopes with agricultural activities. The surface water quality could be improved if serious measures are taken to stop erosion from illegal mining sites and agriculture lands.

Deforestation is one of the most significant environmental consequences of mining, including ASM. In most cases, this is the first major intervention into the natural environment after the start of mining, where deforestation is carried out to prepare land for surface mining [61]. Due to the population growth, which relocates there for mining, there is a higher demand for wood and charcoal in the locality resulting in secondary deforestation. Partow et al. [77] note that the majority of the population is dependent on wood heating and that up to 90% of power produced uses wood and charcoal as a source. According to the Rwanda government, ASM affects deforestation mainly through illegal logging in protected forested areas [18]. Nature conservation areas are an important element in the land conservation efforts [78]. A higher proportion of nature conservation areas is associated with a higher proportion of forest areas [79].

The research studies show that the structure of land use in the whole territory of Rwanda has changed very significantly over the last 50 years; specifically, in the period from 1960 to 2007, the original forest area decreased by 64%. Anthropogenic activities related to mining activities, predominantly ASM and refugee resettlement, had the greatest impact on this rapid decline. On the other hand, the government-level initiatives have resulted in reforestation, which, however, replaces the original forest covers so far to an insufficient extent. Over the last 20 years (more accurate data are from 1990), an average of 2600 ha of forest per year has been reforested in Rwanda [80]. Table 2 shows the loss of forest areas in a total of 18 areas between 1984 and 2015. Illegal gold mining is carried out using the ASM method in some of these areas. According to the Rwandan government report Forest Investment Program for Rwanda, ASM is affecting deforestation mainly through illegal logging in protected forested areas. Illegal ASM degrades large areas of original forest and quality soil through unsustainable logging. With legal ASM, deforestation may occur to a lesser extent. Deforestation usually concerns the felling of trees at the boundaries of the mining area or, secondarily, the illegal felling of trees for fuel. Mining activities are thus connected with illegal deforestation, water pollution and soil contamination [79].

**Table 2.** Rwanda forest area loss between 1984 and 2015 [18,80].

| Location (Forest Area) | Total Area of Forested Areas (in ha) in | | Forest Area Loss 1984–2015 (in %) |
|---|---|---|---|
| | **1984** | **2015** | |
| Buhanda Natural Forest | 1.116 | 18 | 98.4 |
| Gishwati Natural Forest | 21,213 | 1.440 | 93.2 |
| Mashyuza Natural Forest | 85 | 6 | 92.9 |
| Ibanda-Makera Natural Forest | 1.425 | 169 | 88.1 |
| Karama Natural Forest | 3.235 | 1.061 | 67.2 |
| Dutake Natural Forest | 31 | 11 | 64.5 |
| Karehe-Gatuntu Natural Forest Complex | 48 | 19 | 60.4 |
| Nyagasenyi Natural Forest | 45 | 19 | 57.8 |
| Akagera National Park | 267,741 | 112,185 | 58.1 |
| Mukura Natural Forest | 4.376 | 1.988 | 54.6 |
| Sanza Natural Forest | 49 | 24 | 51.0 |
| Mashoza Natural Forest | 36 | 18 | 50.0 |
| Muvumba Natural Forest | 1 286 | 688 | 46.5 |
| Ndoha Natural Forest | 39 | 29 | 25.6 |
| Kibirizi-Muyira Natural Forest | 454 | 352 | 22.5 |
| Busaga Natural Forest | 191 | 159 | 16.8 |
| Nyungwe National Park | 112,230 | 101,005 | 10.0 |
| Volcanoes National Park | 16,128 | 16,004 | 0.8 |
| Total | 429,728 | 235,195 | 54.7 |

## 3. Materials and Methods

### 3.1. Methods of Research

The methodology is identical to the author's published paper in 2019 [19]. The author draws on the qualitative inquiry, which was carried out in accordance with Gerring's [81] conceptualization of qualitative methods as tools for causal inference. At the beginning of the research process, a review of the scientific literature and scientific databases, including the Web of Science was performed, followed by the analysis of data sources and formulation of preliminary conclusions. This study does not consider chemical pollution as an anthropogenic consequence but given the specificity of mining in the locality of interest, it takes into consideration only the anthropogenic impact on the geomorphological forms of the relief. The methodology is based on the collection and analysis of data provided by current authors and research institutions and the research within mining company and field research in 2012–2015. One of the methods of qualitative research was the interpretation of expert interviews, the aim of which was to interpret the views of interviewees on the issues associated with alluvial ASM under investigation [82]. The chosen method was unstructured interviews (according to Hay, 2000) [83] covering key topics in the field of mineral raw material mining with a focus on alluvial ASM. A total of six experts were included in the qualitative research (results from expert's interviews are in Table 3). The technique of their selection was aimed at the maximum possible opinion diversity and representation of key players. These experts were identified, approached and interviewed through snowball sampling and the support of assistant in mining company in Rutsiro district. Methodology of snowball sampling is used in similar research by Bansah et al. [7] in Ghana and McQuilken et al. [84] in Ghana. During the research, open and unstructured research plans were preferred. The task was to create a holistic overview of the researched issue, to capture how the process participants interpret the situation, and to capture the interpretations of these interpretations [18]. Another data collection method was participant observation (according to Clifford et al., 2016 and Kawulich, 2005) of mining sites in the region of interest [85,86]. Results are implemented in the case study. The overall research design draws on the theory-confirming case study by Lijphart, 1971 [87]. The case selection technique was based on the typical case (Seawright and Gerring 2008; [88] for the most recent discussion see George 2019 [89]). Rutsiro district represents a typical example of ASM, and empirical findings from this case study may be generalized for the entire Great Lakes Region.

**Table 3.** Key problems according to experts in field of alluvial ASM.

| Experts (1–6) Answers | Most Mentioned Results from Interviews |
| --- | --- |
| Expert 1, 2, 3, 4, 5 | Example of Rwanda illustrates that the country is more dependent on arable land or its natural resources than other countries in the region. |
| Expert 4, 5, 6 | Due to alluvial ASM, it is sometimes necessary to cut down trees (i.e., illegally) in order to be able to mine in places where miners believe that the mineral vein continues. |
| Expert 3, 4, 5, 6 | The wood obtained due to alluvial ASM is used as a fuel for the food preparation. Due to the felling of trees at the mining site, landslides occur in places where this phenomenon had previously never occurred. |
| Expert 2, 4, 5, 6 | The mining of aggregate material from the riverbed degrades and destroys the present aquatic fauna and flora ecosystems and significantly increases the silt load into the downstream river system. |
| Expert 3, 5, 6 | Sediments in the lower parts of the riverbed, which come from higher places where the mineral washing process takes place, lead to loss of shelter and spawning grounds for fish. |
| Expert 2, 4, 5, 6 | High sedimentation load results in the limited penetration of sunlight into the river system, thereby limiting growth of algae and aquatic plants. |
| Expert 1, 2, 3, 4, 5, 6 | Mining in riverbanks exacerbates erosion, landslides and pollution of water used for sanitary purposes. |
| Expert 1, 2, 4, 5, 6 | Alluvial ASM leads to unmanaged release of tailings into waterways |
| Expert 2, 5, 6 | Smaller streams can stagnate due to numerous open pits and clogging of springs. |
| Expert 1, 3, 6 | Mining areas are close to agricultural land and residential land. Alluvial mining leads to endangering agricultural and residential lands. |
| Expert 3, 5, 6 | Alluvial mining directly endangers the health and property of the local population and damages the environment. |
| Expert 1, 4, 5, 6 | The advantage of mineral extraction is that it is a permanent job, and in the case of a poor harvest, it is the only source of income for the population. |
| Expert 1, 2, 5, 6 | A major problem in environmental protection is non-compliance with the legislation, or inconsistent control by state institutions. |
| Expert 2, 3, 6 | Lack of experts and professional institutions. Professional institutions seek to educate geologists, mining engineers, technicians, and other professionals involved in mining practices that minimize the negative impact of mining on the environment. |
| Expert 1, 2, 4, 6 | Lack of workshops and seminars for miners in the past. Workshops and seminars for miners organized by governmental or non-profit organizations can advise miners on how to approach mining with less undesirable impact. |

expert 1—mining specialist (World Bank), expert 2—owner of mining company, expert 3—geologist, expert 4—government worker (management of national resources), expert 5—academic worker, expert 6—miner.

## 3.2. Study Area

From a geographical standpoint, Rwanda is located at the watershed of two of the most important African rivers—the Nile and the Congo. In the Congo Basin, Rwanda shares water resources with the Democratic Republic of the Congo, through a number of smaller tributaries leading to the free-flowing Kivu Lake and the Rusizi River. Due to its location, it is also known as the "water tower" of the countries in the Nile Basin (Burundi, Democratic Republic of Congo, Egypt, Ethiopia, Kenya, Rwanda, South Sudan, The Sudan, Tanzania, Uganda), which have been cooperating together since 2010 on water resources management within the Nile Basin Countries network [90]. The main source of water supply is atmospheric precipitation, which is, however, spatially very uneven and the volume of which decreases in the direction from west to east The Congo River Basin covers 33% of Rwanda and drains 10% of the water. The Nile Basin covers 67% of the total territory of the state and drains 90% of water resources. Rwanda is covered by a dense hydrographic network consisting of 101 lakes, 861 rivers and 860 wetlands. Although Rwanda is a mountainous state, wetlands cover a total of 10.6% of the territory, of which 53% has been converted to agricultural land and 41% remains covered by natural vegetation and 6% are fallow fields [91]. It is wetlands and swamps that can be endangered by alluvial ASM. Swamp lowlands in the system of deep valleys are used for growing agricultural crops, which are irrigated during the dry season with the help of artificial canals. Wetlands are also the most productive ecosystem in Rwanda and ensure ecological and socio-economic function. They serve not only as a source of drinking water, but also peat, which is used for a fuel. In drought periods, they provide a steady source of water and help regulate floods during heavy rains. Wetland ecosystems help to significantly diversify the landscape and are linked to the habitat of large amount of animal life, which represent a significant proportion of the livelihood for the local population. Groundwater

makes up 86% of the total available drinking water. In the southern and eastern provinces, most of the population is dependent on groundwater, which is transported to the surface by pumps. Although groundwater is considered to be cleaner and of better quality, there has recently been pollution of groundwater resources, mainly due to the use of agricultural fertilizers. This trend is reinforced by the high susceptibility of tropical soils to erosion, with fertilizers penetrating the soil more easily [92].

The Rutsiro district in the Western Province of Rwanda, located 150 km northwest of the capital Kigali, was chosen by the authors for the case study. The Rutsiro district (1157.3 km$^2$) is one of the seven administrative units that account for the Western Province and includes 13 administrative sectors divided into 62 areas and 485 municipalities, which accounts for 3.3% of the total number of municipalities in Rwanda. The Rutsiro district has a population of more than 300,000, which is more than 3% of Rwanda's population. The population density reaches 255 people per one square kilometer. A characteristic feature is a high proportion of the youngest population, where 50% of the population is of pre-productive age, and of which more than 60% of the population is under 25 [93]. The main source of income for the population is agricultural activities and mineral extraction. The agriculture industry serves primarily to fulfill a self-sufficiency function, so the locals are thus significantly dependent on the natural conditions that condition these activities.

The Rutsiro district is a mountainous area located at an average altitude of 2400 m. It is characterized by steep slopes and deeply cut valleys, as deep as 200 m. Due to large total rainfall (average annual precipitation reaches 1200 mm), loose slopes are prone to erosion. The season known as "long rains" is from March to April, during which it falls between 40–60% of the total annual totals. The long rain season alternates with the long dry season, which is between June and September. This is followed by a season of short rains from September to December. From December to March there are some shorter dry periods with prevailing days with no rainfall [94] The Rutsiro district has acquired two mining concessions (Sebeya and Rutsiro) in Rwanda, which are divided between different mining companies. For the needs of this paper, the Rutsiro concession was selected as a case study (Figures 2 and 3).

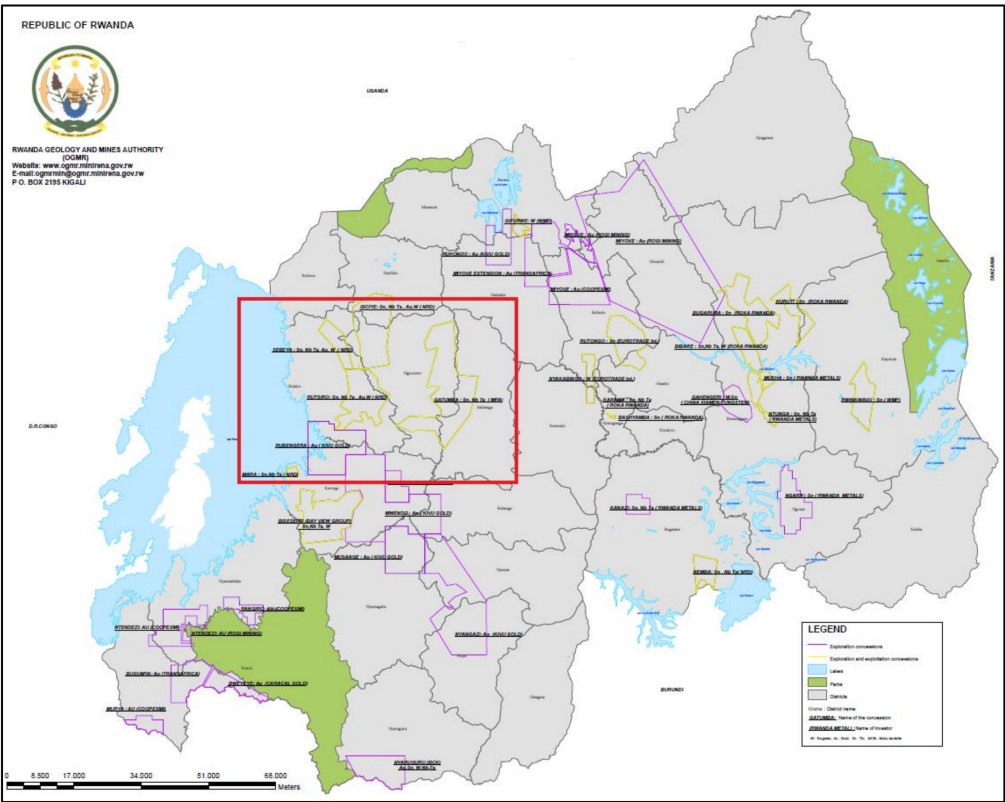

**Figure 2.** Great Exploration and Exploitation Concessions in Rwanda [95].

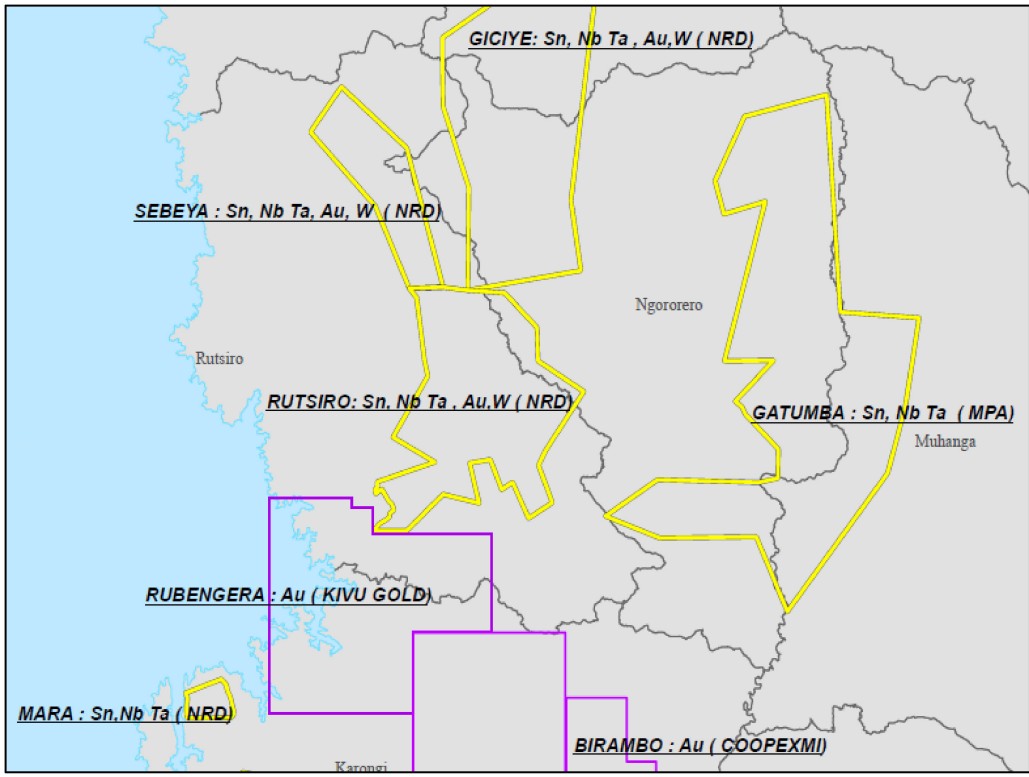

**Figure 3.** Concessions in the northwest Rwanda in detail [95].

The Rutsiro area is located in the zone known as the Central Tungsten belt of Rwanda. Mineral deposits are located in concordant and discordant quartz veins arranged in intermediate quartz and graphitic shale of the Gikoro and Rwandan groups (belonging to the Middle Proterozoic period, 1000–1500 million years ago), which were formed during the Kibaran orogeny. Mineralized structures are oriented in the NW-SE direction. These layers underwent a very low degree of metamorphosis [96]. The northern and southern sections of the mineralized structures are blocked by tectonic faults in the NE-SW direction, which predetermine and tectonically condition the main valley. Mineralization is probably associated with the entry of late Proterozoic to Paleozoic granites (approximately 400 million years ago). Until now, there is no exact stratigraphy for this area, so it is very difficult to define individual strata [97]. In the Rutsiro concession area, water from watercourses is used for drinking and cooking, washing, sanitation, construction and mining industry, and agriculture (irrigation). Drinking and cooking water is collected from springs, wells, and rainwater collection, while other activities, such as agriculture and other economic activities, including the extraction of minerals, use surface water from watercourses and these activities also contribute to their pollution [98]. Artisanal mining negatively affects water quality at the Rutsiro locality, as documented by chemical analysis of samples performed by Haidula et al. [99], implemented in order to prepare an environmental impact assessment. In the mining area, samples were taken from three sites. The collected surface water samples were compared with the East African Standards and further compared with the standards set by the Rwanda Bureau of Statistics. The samples contained excessive amounts of tin, zinc, tantalum, lead, and arsenic.

## 4. Results-Rutsiro District—Case Study

*Anthropogenic Landforms Caused by ASM*

Mining anthropogenic processes are triggered by the extraction of minerals from the earth's crust. The mining landforms can be distinguished into actual and accompanying mining landforms (anthropogenically conditioned forms). The actual landforms are then defined as landforms produced by surface and subsurface mining [74].

Dávid [100] classified the three main groups of landscape damage during mining activities by actually distinguishing three basic groups of anthropogenic landforms:

- Excavated or negative landforms—the most prominent of which are shafts and trenches;
- Accumulated or positive landforms, represented by landfills, the shape of which is determined by several factors, including the earth's surface, the accumulation regime, and the physical properties of the discharged material;
- Areas destroyed by mining, leading to levelling of the surface.

In other words, they can be defined as concave, convex and flat anthropogenic landforms. In the locality of interest, landforms associated with artisanal mining were mapped and inventoried in detail. The emergence of new anthropogenic landforms in connection with the artisanal extraction method is related to significant volumes of mined raw materials, which are mined and then moved while part of the waste material is stored. In areas with ASM, new landforms are created, most of them by anthropogenic activities. The most common landforms include mines, pits, adits, hollows, and fluviatile placers.

New anthropogenic landforms are formed in the wide riverbed and alluvial plain (in fluvial sediments), where the next phase of water-transported material extraction takes place. Lighter and smaller metal minerals, which were not mined in an upper stream, are transported as floating solids and accumulated in the valley floors of a lower stream.

In a place with a wide riverbed and where the river is not so deep, there is a secondary extraction of minerals from deposited sediments. Miners extract coarse-grained deposited matters from the bottom of a riverbed and from banks, which they sluice using mining pans. This process helps to form fluviatile placers and fluviatile placer fields which are of convex shape. Fluviatile placers are anthropogenic forms of relief created during panning, i.e., the mechanical method of mining from alluvium, most often during the mining of gold and other pure metals or gems, or perhaps moldavites and pyrites. They are small accumulation heaps of gravel and sand, which are an accompanying landform of the metal panning method. These are mounds usually 1–2 m high, in rare cases higher than 10 m, often in one locality with very different heights. If they occur on larger areas, they are referred to as fluviatile placer fields. They frequently occur in the alluvial plain, and in artisanal mining they can more significantly interfere with a lower part of the slope (Figure 4).

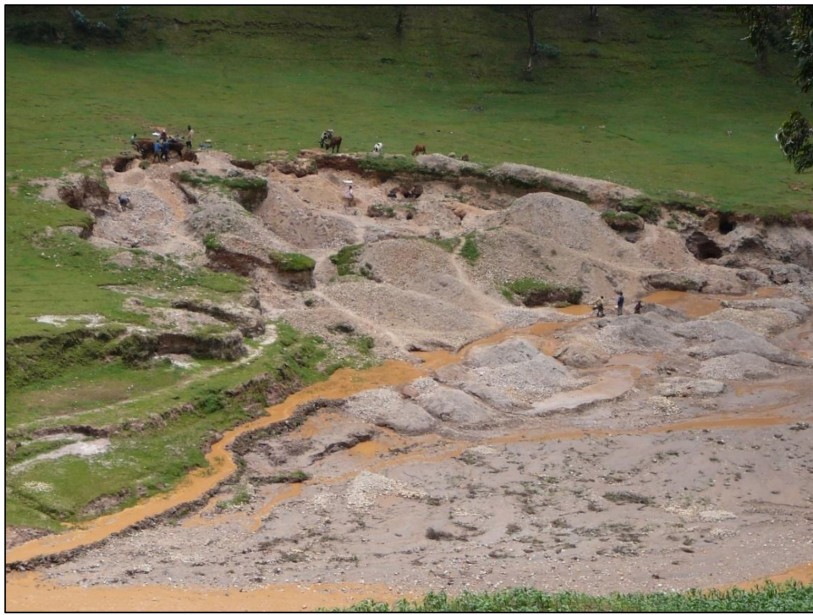

**Figure 4.** Fluviatile placers and fluviatile placer fields. Photo taken by the author [18].

Another anthropogenic landform is barren rocks and their influence on the lower part of the slope of an alluvial plain disturbed by ASM (Figure 5). The barren rock generated by this activity are in many cases commonly discharged and moved to the surroundings in the mining area due to insufficient legislative standards or simply due to a lack of storage facilities. Therefore, these technological processes have a great influence on geomorphological processes and on the environment [69]. The actual geomorphological processes also take place on heaps, in barren rocks, or in areas where wastewater is stored. The deposition of barren rock and the consequent instability of slopes leads to the acceleration of slope processes [101]. Then, the extracted material can enter a watercourse and lead to an increase in sediments and mud in the landscape, despite the transfer of toxic material downstream and subsequent deposition elsewhere downstream [39,102–105].

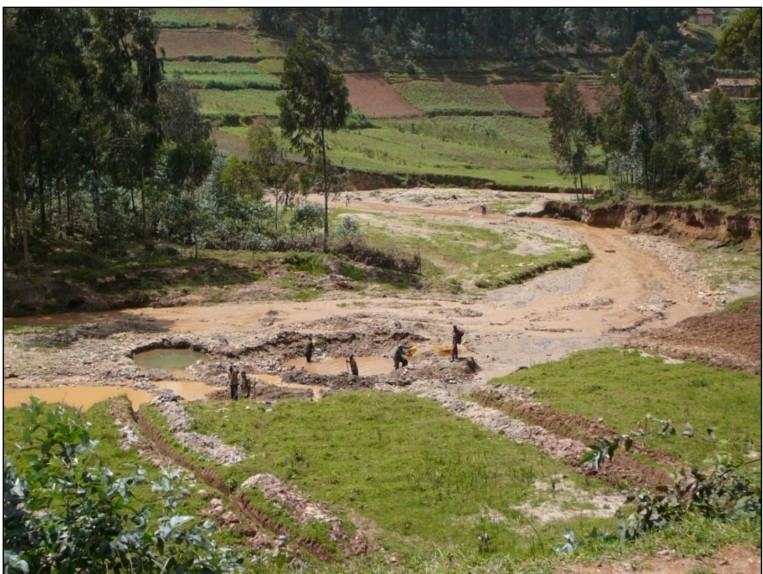

**Figure 5.** Influence on the lower part of the slope of an alluvial plain disturbed by artisanal and small-scale mining. Photo taken by the author.

During mining activities in the riverbed, there is an anthropogenic influence of the alluvial plain by erosion and accumulation processes. During the material transfer and the barren rock formation, the watercourse branches and gravel benches are formed. The riverbed is affected by lateral erosion, with stream turns and bank ruptures occurring (Figure 6).

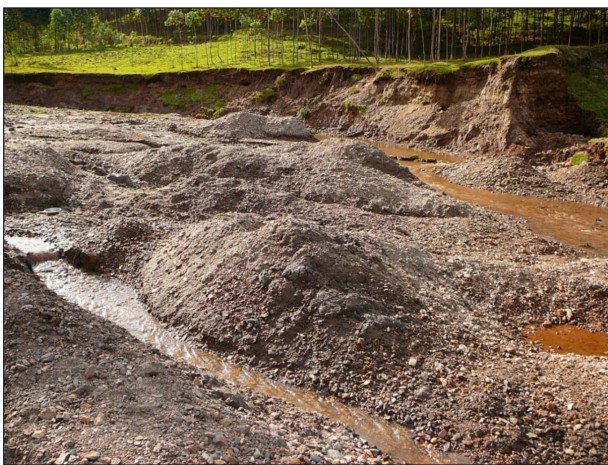

**Figure 6.** Watercourse branches and gravel benches. Photo taken by the author.

Lateral erosion is then followed by intense deep erosion in the alluvial plain, which leads to a deepening of the watercourse bed (Figure 7).

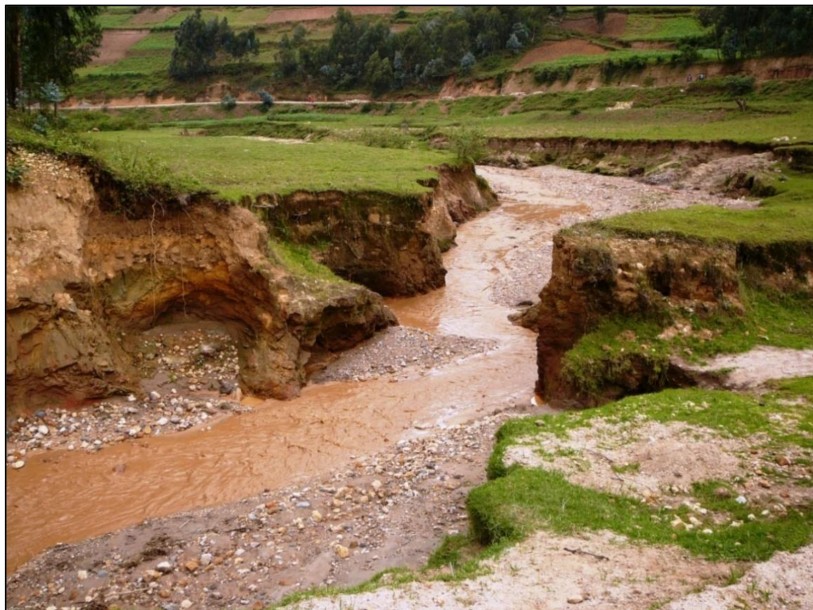

**Figure 7.** Erosion in watercourse bed. Photo taken by the author.

Fluvial processes create anthropogenically conditioned ravines. The ravines reach a depth of up to 10 m, are actively modelled by deep erosion and are often formed on deforested areas, meadows, and pastures (Figure 8).

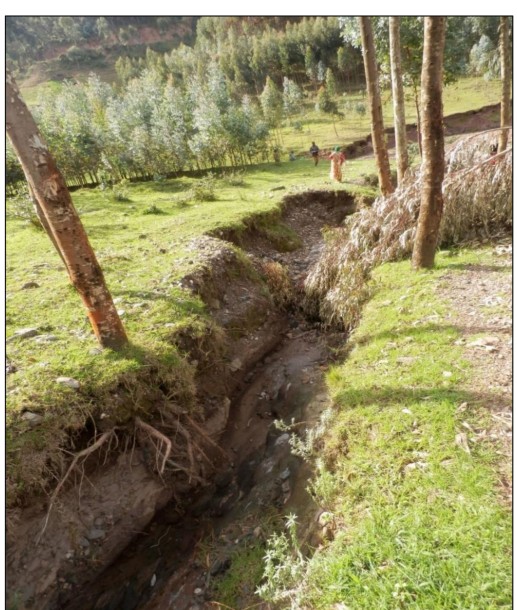

**Figure 8.** Ravine. Photo taken by the author.

In the areas with a disturbed surface because of artisanal mining, landslides and rock block slides occur. For the middle parts of the slopes located directly above the watercourse bed, there are valley slopes with clear separating surfaces disturbed by the slope processes (Figure 9). Deep erosion causes the expansion of ravines and the removal of part of the material by subsurface selective removal, which results in the subsidence of the surface.

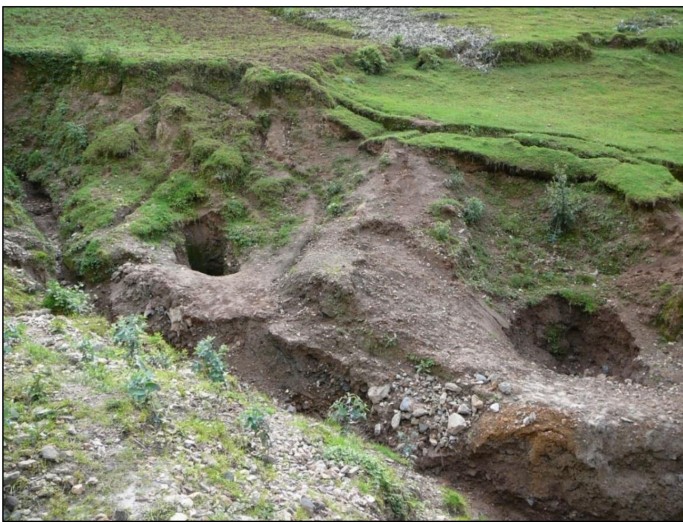

**Figure 9.** Slope processes. Photo taken by the author.

The water outflow in a mining area itself is fundamentally affected. The new landforms are artificial canals and dams of small water works, which are built with the aim of provision of a sufficient water source for sluicing the ore by a panning method. After washing the ore, the water course will be dammed in a higher position and the water will be diverted to another mining site. Accumulated minerals are collected at the washing site and mining can continue. Due to the need for the use of flowing water, many artificial canals with a system of weirs and dams are created in the quarry site, which are constantly disturbed along their entire length by some mining technology, but also by natural deep (ravine) erosion during daily heavy rains. As a result, there are areas deeply divided by ravine erosion with individual ravines merging into very deep fluvial erosion landforms, which subsequently lead to slope processes.

The anthropogenic landform, which is to prevent erosion and partial deposition of sediments, are the so-called check dams. Check dams reduce the effective slope and create small pools in swales and ditches that drain 5 ha or less. Reduced slopes reduce the velocity of storm water flows, thus reducing erosion of the swale or ditch and promoting sedimentation. The use of check dams for sedimentation results in the net removal of sediments. Use of a series of check dams will generally increase their effectiveness. A sediment trap may be placed immediately upstream of the check dam to increase sediment removal efficiency (Figure 10).

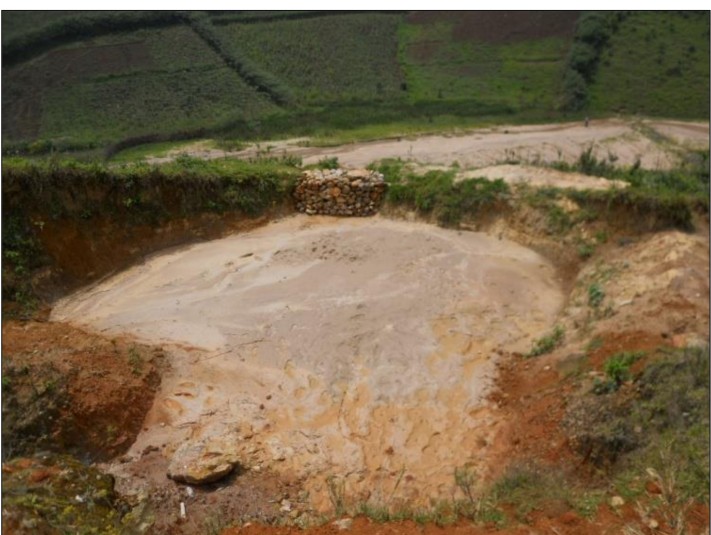

**Figure 10.** Check dam. Photo taken by the author.

The convex form is then surface depressions. These are unintended terrain depressions created by rapid settling, sinking or collapse of mine works. The ground plan of surface depressions is most often circular in the area (above the intersections of the mine tunnels) or elliptical (created by connecting two circular surface depressions). Circular surface depressions usually have a diameter of up to 10 m and a depth of 3 to 5 m. They are sometimes filled with water, but unlike sunken areas, there is usually no permanent year-round water level in them. Suffosion depressions are formed in the alluvium areas, when adits are built in bank ruptures, the ceiling of which collapses in soft rock (Figure 11).

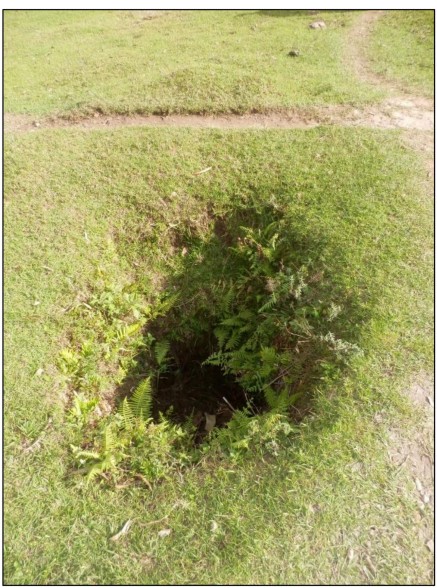

**Figure 11.** Suffosion depression. Photo taken by the author.

Loose banks are more prone to lateral erosion, which is manifested along the entire length of a river by the formation of bank ruptures and landslides. Thus, fluvial processes in particular are significantly affected. Slope, rill, sheet and gully erosion processes as well as volume extension of suspended material, which are conditioned by mining activities, especially slope processes (landslides and collapse) and land subsidence in undermined areas were also documented in the area of interest. Artisanal miners disturb unstable exposed ground. Their mining techniques contribute to choke of lower part of water streams in riverbeds, valley floors and riverine plains (Figure 12).

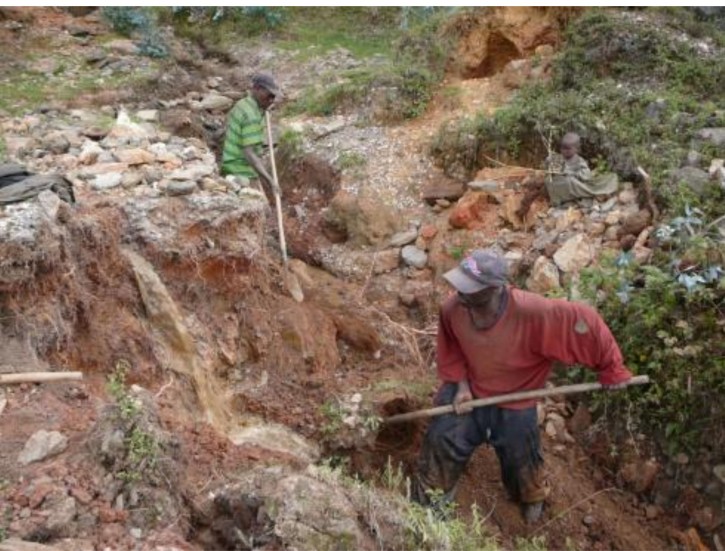

**Figure 12.** Artisanal miners disturb unstable exposed ground. Photo taken by the author.

On the valley slopes, due to deforestation, risky mass movements occur (Figure 13) when the material from the slope is transported to an alluvial plain by erosion processes and landslides. In the alluvial plain, in the area of a wider valley floor, there are fluviatile placer hills as a remnant of secondary mineral mining from deposited sediments. These are small accumulation heaps of gravel and sand (1–2 m high), which are an accompanying form of a panning method.

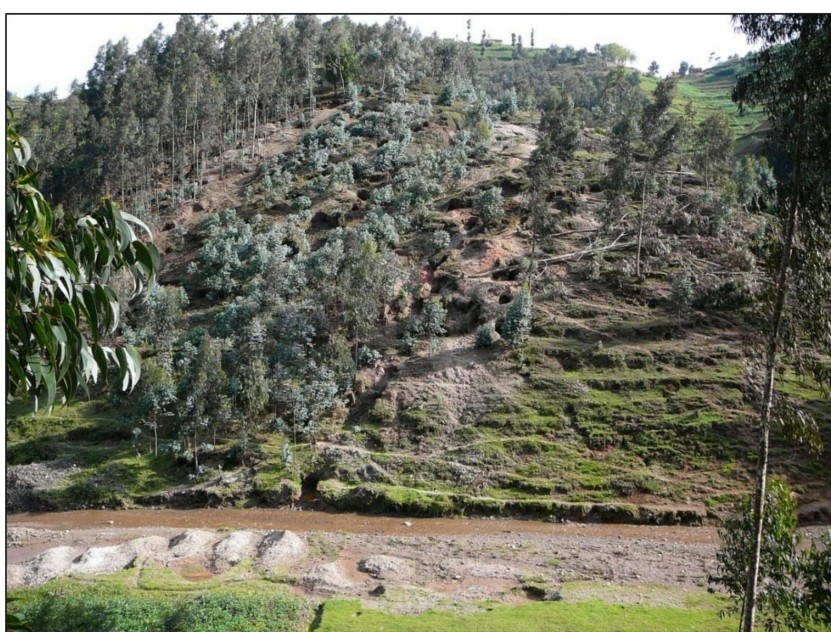

**Figure 13.** Deforestation. Photo taken by the author [18].

Due to lateral and deep erosion during mineral extraction, watercourses are widened at the expense of agricultural land and forest remnants. Although logging is banned due to limited resources, deforestation occurs when mining is expanded. It does not concern deforestation of large areas, but rather the destruction of individual trees along the watercourse and the mining site. If a miner follows a mineral vein located under trees, he simply either cuts down the tree or, undermines it in the best-case scenario. These procedures destroy a root system and lead to a loss of channel water. These activities have a negative impact on both groundwater resources and the entire water cycle.

Alluvial ASM leads to acceleration of natural geomorphological processes and significantly affects a fluvial landscape. Alluvial ASM, as a specific part of the ASM topic, has several environmental aspects. According to the created typology [18], the greatest impact of alluvial ASM involves naturally occurring geomorphological processes, such as weathering, mass movements, fluvial processes and the creation of new anthropogenic forms. In terms of changes in landscape structures, ASM mostly affects primary and secondary deforestation and the associated land cover change. Alluvial ASM significantly affects the hydrological regime by water contamination and sedimentation of water streams. Alluvial ASM has an effect on soil fertility by contaminating soil that can no longer be used effectively to grow crops. The mining of aggregate material from the riverbed degrades and destroys the present aquatic fauna and flora ecosystems and significantly increases the silt load into the downstream river system. High sedimentation load results in the limited penetration of sunlight into the river system, thereby limiting growth of algae and aquatic plants. High levels of silt further suffocate the river systems and impacts on the spawning of fish. The mining of the riverbed further leads to physical disruption of the hydraulic characteristics of the river itself.

The findings emerged from the interviews with main stakeholders (representatives of the mining sector, research institutions, municipalities, miners, local inhabitants, etc.) in the Rutsiro District are shows in Table 3.

Mining activities in the Great Lakes Region are viewed as an opportunity to raise funds for the region's development. Governments are trying to combine two approaches in the use and protection of the environment. The first approach is to use natural resources as effectively as possible for economic growth. The second approach, on the other hand, is to protect the region's natural resources and not irreversibly damage the environment. Experts use the example of Rwanda to illustrate that the country is more dependent on arable land or its natural resources than other countries in the region. The advantage of mineral extraction is that it is a permanent job, and in the case of a poor harvest, it is the only source of income for the population. Due to ASM, it is sometimes necessary to cut down trees (i.e., illegally) in order to be able to mine in places where miners believe that the mineral vein continues. The wood obtained in this way is then used as a fuel for the food preparation. It was further confirmed that due to the felling of trees at the mining site, landslides occur in places where this phenomenon had previously never occurred. At the same time, however, they claim that wood is very expensive and is needed for food preparation, so in the case of tree felling, the wood is divided among other miners. A major problem is water pollution in the river channel. Mining areas are close to agricultural land and residential land. Mining in riverbanks exacerbates erosion, landslides and pollution of water used for sanitary purposes. Mining thus directly endangers the health and property of the local population and damages the environment. Sediments in the lower parts of the riverbed, which come from higher places where the mineral washing process takes place, are also a problem. Professional institutions seek to educate geologists, mining engineers, technicians, and other professionals involved in mining practices that minimize the negative impact of mining on the environment. However, this process is slow and the interest in experts is great due to the developing mining industry. The problem is illegal mining activities that take place in mountainous remote areas. Although there is a local government in these areas, it is often associated with illegal activities and some officials support illegal mining. Part of the profit from smugglers or illegal mineral traders then comes from their support or at least through them turning a blind eye.

## 5. Conclusions

Primary ASM takes place on the slopes of the mining site and then in the spring sections of watercourses and their tributaries, where the naturally high flow power is utilized due to gravity. Minerals are mined in narrow pits and adits, many as deep as several dozens of meters. The ore-bearing rock is transported to a watercourse, where it is subsequently sluiced by the panning technique. The mined minerals (coltan, tin, tungsten, tantalum) are heavier than ordinary rocks, and thus sink to the riverbed, where they are retained. After sluicing the mined rock, the river is dammed and water is diverted elsewhere. During this method of extraction, only the extraction of minerals of larger sizes is allowed, and therefore large losses occur. Due to the use of flowing water, many artificial canals with a system of dams, which are constantly disrupted along their entire length, are created in the mining area. Disruption occurs as a result of mining techniques and natural deep (ravine) erosion during daily rainfalls. The next level of mineral extraction is mining in a wide riverbed and in fluvial sediments. Lighter and smaller metal minerals, which were not mined upstream, are transported as floating solids and accumulated in the valley floors downstream. In a place where the riverbed is wide and the river is not so deep, a secondary extraction of minerals from deposited sediments takes place [18].

Coarse-grained alluvium is then taken from the bottom of the riverbed and banks, and sluiced with a pan. Loose banks tend to be more prone to lateral erosion, which is manifested along the entire length of the stream by the formation of bank ruptures and landslides. Processes influenced by mining activities lead to mass movements such as creeping, sliding, run-off, and falling. Landslides along cylindrical shear surfaces are significant, which occur mainly in unconsolidated or partially consolidated (in clays and marls, claystones and clay slates) rocks. The separating landslide area then typically has a concave shape, and the landslide masses accumulate at the lower part of the slope. Transverse cracks also form on the landslide, with water accumulating in here, which worsens the

equilibrium conditions of the slope. The collapsed rock is often saturated with water so that the slide has a character of a ground current [106].

Mining in rivers directly influences the canal geometry and causes diversion of the river away from the original canal, sediment accumulation and the formation of deep pits [107]. Mining in alluvial sediments leads to the removal of large coarse-grained materials, stone blocks, and other material that is carried by the stream from the higher sections of the upper stream (parts of trees, branches, etc.). Downstream, sediments accumulate, and chemicals, which are used in the processing of minerals, settle. In addition to the movement and deposition of sediments, mining in the riverbed has an effect on the flow and direction of the river, which then influences the fauna and flora in the river and its lower parts. If the riverbed is widened during alluvial mining at the expense of agricultural land, eutrophication and chemical pollution of areas further downstream occurs.

ASM intensifies lateral and deep erosion. The movement of miners at the mining site, on steep slopes, the actual mining and the sluicing of minerals disrupt the slopes or riverbanks, which then leads to erosion. In the case of mineral sluicing, there is mainly deep erosion and deepening and widening of artificial canals. Deep erosion continues until it reaches a solid subsoil formed by, for example, rock blocks. Lateral erosion mainly concerns mining in alluvium. Erosion and landslides are also affected by the movement of miners themselves, who move the rock from shafts to a mineral treatment site. During this transport, the movement of miners disturbs the unstable rock environment, which in most cases is located on steep slopes. Together with daily precipitations, geomorphological processes are intensified. ASM in the alluvium thus has a significant effect on both the near and far surroundings of the mining area and on the watercourse itself that flows through the site. The author agrees with Byizigiro [58] and Nelson and Church [68], who, in the typology of environmental impacts, place emphasis on taking into account the different types of ASM and their specifics.

To mitigate negative impacts of alluvial ASM, it is recommended that the local residents be encouraged to reprocess the mine's very low grade rock dumps instead of the sand, cobbles of quartz and quartzite located in the river, were possible. This will then provide the community with an environmentally friendly alternative source of rock material and reduce the amount of waste rock presently retained at the mining sites. However, strict safety controls must be implemented. Sand and rock extraction in the rivers must continue to be monitored and prevented, where possible.

The most important step to mitigate the environmental impacts of alluvial ASM is strict compliance with rules and laws and greater control of mining activities on site. At the same time, miners need to be trained in sustainable mining practices, and reduce illegal mining.

Environmental impacts of alluvial ASM are much more significant in rural areas where the population is vitally dependent on agricultural land and water resources. Due to the expansion of mining sites, the share of agricultural land is decreasing, and water quality is deteriorating. Further population growth, due to its demographic behavior, will lead to higher demands on land, water, and other natural resources. Without changing the attitude of all stakeholders, environmental impact will cause deterioration. Good practice in mineral mining is a necessary condition for a sustainable approach to the management of natural resources and improving the well-being of the local population.

This study provides readers with an introduction to the environmental impacts of alluvial ASM in Rwanda. In view of the above facts, the research will help to better understand the need for sustainability of mining operations and may contribute to better environmental protection and improving of well-being in local communities. This study may also be interesting for international journal readers.

**Funding:** This research was funded by University of Ostrava, grant number SGS11/PřF/2020-Smart cities and other innovative approaches to urban and regional development.

**Acknowledgments:** The author gratefully acknowledges helpful comments on earlier drafts from Veronika Kapustová, Ondřej Slach and Jan Ženka.

**Conflicts of Interest:** The author declares no conflict of interest.

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
