# Peer review of "Alluvial Artisanal and Small-Scale Mining in A River Stream—Rutsiro Case Study (Rwanda)"

_forests, doi:10.3390/f11070762_

Round 1
Reviewer 1 Report
The article is merely more than a description of a trip to Africa It is not a scientific article, there are no real methods (just visiting sites), thus there are no results.
It is quite far from the requirements of a scientific paper. In the present form, it must be rejected without any hesitation.
The numbers bellow refer to the numbered lines.
Title: It is very unfortunate to have abbreviation in the title. What does ASM refer to?
Abstract
In the abstract, please, be more specific about your results. The last sentence is just a repetition of the previous ones. Please, consider it to delete, and extend the real results of the paper.
L.16: “ to increasing and reshaping of soil shapes” What do you mean on soil shape?
Introduction
The first paragraph of the introduction must be organised more clearly: e.g. not to jump from chemical changes to morphological ones, and later again to chemical problems. Please re-arrange the entire paragraph in a logical order. Besides, you should give a literature review here, so the reader could see the unsolved questions in the topic, and see whether it is just a case study, or the results could be broadly applied. This literature review is missing.THe
20: I would delete the „among other things” and use terminus technicus for the extracting, not just „to flush out minerals.”
24” Given the presence of minerals in the alluviums, ASM has an impact on the riverbed.” Geomorphologically the river and the floodplain are two different units, but here you mix them. If there is mining activity on the floodplain, how could it affect the riverbed?
31: „removal of a large amount of rock further downstream„ I do not understand it, please, explain the mechanisms!
47: “Great Lakes Region (GLR)” if you use the abbreviation only twice, consider not to use it! Plus, specify the country!!
48: „[20–32].” There are thousands of references in this topic. Why did you select the given ones? How do you evaluate them? I do not think it is appropriate just to cite a dozen article without any real evaluation, just to make your reference list longer.
Why did you highlight mercury, if the pollution in your area is caused by other minerals?
At the end of the chapter please specify your exact and detailed aims!
2.1. Study area
As the reader might not have as wide knowledge on Rwanda’s geography as you, please, create a map where ALL the names you list (thus you think it important) are indicated. If they are not important, do not mention them.
Here the structure is also poor, as for example lakes or wetlands pop up several times, independently of each other.
The chapter is too long, the readers are not really interested in the general geography of the country. Fort hat they can search the internet…
63-64: „at the confluence of the Nile and the Congo”. Are you sure? It is the greatest geographical mistake I had ever seen in an article. Confluence means: when two rivers join. Is it the case?
73: „percent” please use %
81: „53% has been converted to agricultural land and 41% remains covered by natural vegetation” What about the remaining 6%?
82: Such a list like “the Nyabarongo, Akagera, Base, Muvumba, Koko and Rusizi Rivers. The largest lakes include Kivu Lake, while other two Cyohoha and Rweru Lakes” are totally useless. It is like a topography class, I am sorry to tell.
100 „The speed of groundwater movement is estimated at 66 m³/s”. I do not understand this, plus the flow rate probably influenced by several factors.
Where is you EXACT study area? Please, describe that part and give a figure on that HERE.
2.2. Methods of research
107-112: Do not write such a general sentences, they do not fit to a scientific journal.
112: „The primary objective..” Aims should not be mixed into this chapter.
112” anthropogenic consequences” I do not understand this expression: how could a man-made process (=mining) have athropogenic consequences?
The whole section is just a mixed-up mentioning of general methods. Within the study you publish in this article, why methods did you apply? What was the data source? What did you measure? What was the error of these measurements? Just to visit places is not a scientific research, I am sorry to tell you.
As far as the methods are not described clearly, the results can not be evaluated.
3.1. Human impact of geomorphological processes of ASM
129: 3.1. Human impact on?? geomorphological processes connected to?? ASM
Results
These are not really the own results of the author, just general statements. It is very unusual, that comparison and literature citations are mixed with the supposedly own results.
Actually, the reviewer can not evaluate this chapter, as the methods were not described clearly, and here no real measurements are introduced.
The 3.1. chapter is more of an introduction, that a result chapter, especially that it is based on a reference, and not on a case study.
The 3.2. Chapter has also nothing to do with results, it is a study area description.
- 301. Where is exactly the Rutsiro district? Please, provide a map, + the area description (L. 302-354) should go to the Study area chapter.
- 313 „in Table 20” Such citations make me afraid of the possibility of plagiarism. The Editor must check it, before accepting the paper.
Chapter.3.3.
The author just made some pictures, and give a general descriptions to these areas, not even closely related to the site. No measurements were made.
Author Response
Dear reviewer,
Thank you very much for your suggesting comments on my manuscript. I have incorporated all your suggestions into the text. I am very sorry about some unnecessary mistakes that my poor English translation caused. Thanks to your remarks on the text structure, revision of some information and refinement of terms, the paper content has significantly changed. I have now attached the file version with track changes. I have also attached a Word document, where I commented on your individual remarks.
Again, thank you very much for your comments and input.

Reviewer 2 Report
The problem of the impact of artisanal and small-scale mining (ASM) on land relief in Rwanda is very interesting and important. After reading the manuscript, however, I have some critical comments.
First of all, I doubt that Forests journal is the best place to publish this article. This is due to the fact that the main reason for changing the relief is mining. Deforestation is mentioned only in some places in the text. We do not know how many areas have been deforested because there is no information about it in the text. The analysis of the impact of deforestation on fluvial processes is more complicated than that described in the manuscript. This not only influence on the evaporation rate. This also affects groundwater, for example. These relationships have been described in many scientific articles (eg. Falkowski, 1975).
This is a partially review article (chapter 3.1). Only chapter 3.2 describes the author's own results and observations, but also not in a complete way. In the Methodology chapter we can find information that one of the research methods were "interviews covering key topics in the field of mineral extraction, with particular regard to alluvial", and that a total of six experts took part in qualitative research. The specific results of these studies are not in the manuscript. This should be presented in the text or in the tables.
What does 'non-participant observation of mining sites' mean?
There is no figure with location of the analysed areas.
The introduction is too long and contains unnecessary information unrelated to the main topic of the work (e.g. General characteristics of the region). Similarly, the introductory part of chapter 3.2 (information on population density, municipalities, soils is not needed). No reference to other regions and to the results of other researchers regarding this problem.
Author Response

(The authors gave the same response as above.)

Round 2
Reviewer 1 Report
Dear Authors,
I still stand by my previous recommendation (e.g. to be rejected), as the research methods are simply just not real scientific methods, and I can not accept answers in Czech(?) see attachment: reviewer1_en.xls.
I wish you all the best!
Author Response
Dear reviewer,
Thank you very much for your suggesting comments on my manuscript. I have incorporated all your suggestions into the text. I am very sorry about some unnecessary mistakes that my poor English translation caused. Thanks to your remarks on the text structure, revision of some information and refinement of terms, the paper content has significantly changed. I am sorry for previous xls file where was my notes. During, quite stressful situation I attached wrong file (xls. file instead of doc. file). I explained my methods and data collection. I have also attached a Word document, where I commented on your individual remarks.
Again, thank you very much for your comments and input.

Reviewer 2 Report
There are still no interview results presented. Information from specialists should be shown e.g. in a table. The title of Chapter 2 does not correspond to the content. This is too general. Content concerns the anthropogenic impact of ASM on geomorphological processes.Author Response
Dear reviewer,
Thank you very much for your suggesting comments on my manuscript. I have incorporated all your suggestions into the text. Thanks to your remarks on the text structure, revision of some information and refinement of terms, the paper content has significantly changed. I added table 3 with results from interviews with experts in field of alluvial ASM. I have also attached a Word document, where I commented on your individual remarks.
Again, thank you very much for your comments and input.
